# Sustainable Tourism of Important Plant Areas (IPAs)—A Case of Three Protected Areas of Vojvodina Province

**Igor Trišić** [1,*], **Danka Milojković** [2], **Vladica Ristić** [3], **Florin Nechita** [4], **Marija Maksin** [5], **Snežana Štetić** [6,7] **and Adina Nicoleta Candrea** [8]

1. Faculty of Geography, University of Belgrade, Studentski Trg 3/III, 11000 Belgrade, Serbia
2. Singidunum University, Danijelova 32, 11000 Belgrade, Serbia; dmilojkovic@singidunum.ac.rs
3. Faculty of Applied Ecology "Futura," Metropolitan University, Požeška 83, 11030 Belgrade, Serbia; vladica.ristic@futura.edu.rs
4. Department of Social and Communication Sciences, Transilvania University of Brasov, 29, Eroilor Bd., 500036 Brasov, Romania; florin.nechita@unitbv.ro
5. Institute of Architecture and Urban & Spatial Planning of Serbia, Bulevar Kralja Aleksandra 73/II, 11000 Belgrade, Serbia; maja@iaus.ac.rs
6. The College of Tourism Belgrade, Bulevar Zorana Đinđića 152a, 11070 Belgrade, Serbia; snezana.stetic@gmail.com
7. Balkan Network of Tourism Experts, 11000 Belgrade, Serbia
8. Faculty of Economic Sciences and Business Administration, Transilvania University of Brașov, 500036 Brașov, Romania; adina.candrea@unitbv.ro
* Correspondence: trisici@hotmail.com; Tel.: +381-64-143-13-75

**Abstract:** In the northern part of Serbia, where the Autonomous Province (AP) of Vojvodina is located, a total of 27 important plant areas (IPAs) have been established. Within three selected IPAs, there are different biocenoses inhabited by endemic plant and animal species, which form a unique sensitive ecosystem. Furthermore, in these areas, there are wetlands and areas important for geodiversity maintenance. The three studied IPAs include the territory of special nature reserves of the same name: the Special Nature Reserve (SNR) Zasavica, the SNR Obedska Bara, and the SNR Koviljsko-Petrovaradinski Rit. In these selected protected areas, tourist activities are carried out at different levels of development and through various forms of tourism. This research used quantitative methodology to examine the degree of sustainable tourism development and its impact on residents. A total of 1134 respondents were surveyed using a questionnaire as an instrument. The obtained results indicate that the selected IPAs can be destinations for sustainable tourism development. Ecological and sociocultural dimensions of sustainability contribute to this type of tourism to the greatest extent. Analyzing the data related to the impact of sustainable tourism on the respondents' satisfaction, we concluded that they feel a certain satisfaction with sustainable tourism in all three investigated protected areas. The survey results called attention to the possibility of developing nature-based forms of tourism, in which the residents' role in tourism planning and development should be strengthened. All forms of tourism that are developed here should have the primary goal of protecting nature in these IPAs.

**Keywords:** sustainable tourism; IPAs; protected area; wetlands; nature-based tourism; ecotourism

## 1. Introduction

In the northern part of Serbia (the territory of AP of Vojvodina), there are 27 important plant areas (IPAs), with a total area of 328,208 ha, which represents 15.3% of the territory of Vojvodina [1]. A large part of these IPAs is constituted within various protected areas, such as special reserves and nature parks.

The IPAs Obedska Bara, Zasavica, and Koviljsko-Petrovaradinski Rit are integral parts of the Special Nature Reserves with the same name. Various biocenoses with endemic plant species have grown there, for which they were granted the international protection status

of IPA. In addition, the main characteristic of these areas is wetlands inhabited by very rare animal species and plants, which together form a unique and sensitive ecosystem.

Protected areas contribute to the preservation of biological and landscape diversity, the preservation of ecosystems and specific habitats of wild plant and animal species. In the process of protection, it is necessary to study all activities, analyze impacts and possible changes, isolate existing and strengthen new benefits, and determine how certain measures will minimize negative conflicts, which will result in a tourism destination being formed as a final product. The studied areas were protected in the mid-nineties of the twentieth century. Until then, there were no special protection measures, but the local population was guided by the "good host" principle. After the declaration of these areas as Category IV—Habitat and Species Management Area, the organization of protection and management was approached. The population living around these protected areas belongs to different ethnic communities and possesses significant ethno-social values. Culture and cultural and historical heritage, customs, gastronomy, home craftsmanship, and events stand out among the most important values [2]. The mentioned elements of culture represent essential social characteristics. The natural and social features of these protected IPAs can be important tourist motives that create a destination for sustainable tourism development. The inclusion of residents and visitors in tourism planning and development, the availability of local products, the realization of interaction between residents and visitors, and the involvement of residents in monitoring and protection can produce environmental, economic, sociocultural, and institutional benefits, which are the main objectives of sustainable tourism development [3,4]. Proper tourism development in these IPAs could condition significant local economic development of these parts of Vojvodina. Properly planned tourism development can have the following results: more significant employment of residents, promotion and sale of local products, development of local brands and companies, strengthening and expanding interest in local events, etc. [5]. Along with intensifying social potential, significant investments should be made in protecting natural values [3,6–8].

The research subject in this paper is the examination of sustainable tourism and its impact on the residents of three selected IPAs in Vojvodina. In addition, the role of the local population in tourism planning and development is examined [9]. Besides visitors, the role of residents in sustainable development is of great importance [10].

The objective of this paper is to determine the degree and importance of sustainable tourism within three selected IPAs, with the help of the obtained research results, i.e., to evaluate the impact of sustainable tourism on the residents' satisfaction. In addition, it is crucial to examine four dimensions of sustainability: environmental, economic, sociocultural, and institutional. After collecting the answers from the respondents, it is also valuable to establish the individual impacts of these dimensions of sustainability on tourism development.

The originality of this research is due to the focus on three protected areas of the Srem geographical unit in Vojvodina. In these three selected protected areas, the impact of sustainable tourism through the four dimensions of sustainability (ecological, economic, sociocultural and institutional) has not been examined until now. In addition, the impact of sustainable tourism on residents' satisfaction is being investigated for the first time in this area. The results of those studies indicate a significant impact of sustainable tourism on the satisfaction of residents. Additionally, the previous results specify that the ecological and sociocultural dimensions of sustainability contribute to the greatest extent to sustainability in the examined protected areas. This research is considered a continuation of the author's previous research. It is planned to examine sustainable tourism in as many protected areas as possible that make up the territorial unit of Vojvodina. Through a comparative analysis of the obtained results, the authors compare the overall results, which can help identify weaknesses and opportunities for the development of sustainable forms of tourism, which protected areas can implement in the tourist offer.

The natural values of these IPAs with great biological and ecological importance are the main motives for visitors. The most significant tourism values of IPAs are flora, ornithofauna, and wetlands as primary resources. Accordingly, the forms of tourism that should be primary in these reserves are scientific research and ecotourism, with the basic activities of research and observation of nature, plants, birds, and animals. In addition to the ecological aspect, sociocultural, economic, and institutional aspects of sustainability are also important [10,11]. A destination where the improvement in the protection and values of a reserve is directly supported by the local community, the state, and its authorities, represents an essential basis for sustainable tourism development [12–14].

Compared to the period of the twentieth century, nowadays the protection of natural areas has contributed to the fact that the processes of destruction of natural habitats, landscapes and organic species in them take place more slowly than outside the protection limits. The goal is to stop the destruction and degradation of the research IPA that remains for the near future and beyond. Along with the protection of the area, the local population's awareness that destinations with rare and sensitive elements should be protected is also strengthening. A special quality in the preservation of protected areas is the inclusion of stakeholders from the local community and beyond, which will be investigated separately in future work.

## 2. Literature Review

The selected IPAs are inhabited by especially relict and endangered plant species that must be protected. These rare species can be directly threatened due to improper tourism development in these protected areas. With the absence of a protection regime and the control of application measures, the area can change its original characteristics, adapting to tourists' needs, because wetlands are the areas most susceptible to tourist impacts [15]. Models of space degradation by users can be an increased number of tourists per a particular area in an appointed time interval, non-renewable use of reserve resources [16], poaching, dumping of garbage and waste, the devastation of forest areas, presence of domestic animals, proximity to agricultural land, proximity to ecological pollutants, unplanned tourism development, and other activities. In these protected areas, it is necessary to develop forms of tourism based on the improvement of natural resources, such as ecotourism, scientific research tourism, bird watching, excursions, and other forms of tourism based on the preservation and improvement of natural values. Sustainable tourism can represent a means of tourism protection and development [12,17].

For numerous managers, sustainable tourism in protected areas means planning various measures and their implementation in protection systems [18]. These activities have priority in improving area protection [19,20]. Furthermore, important activities in tourism development must be aimed at the satisfaction of tourists and residents through strengthening mutual interaction [21], local culture, and the economy through implementing various tourist activities [17,22].

Asmelash and Kumar [23] state that the examination of sustainable tourism in protected areas can be conducted by observing and measuring the value and impact of four sustainability dimensions on sustainable tourism and visitors, namely, ecological, economic, sociocultural, and institutional ones. In their research, four hypotheses were set. Hypothesis H1—that there is a direct relationship between the ecological dimension and tourist satisfaction—was not confirmed. However, hypotheses H2, H3, and H4 were confirmed, and they concern the establishment of a direct relationship between the economic, sociocultural, and institutional dimensions of sustainability and visitors' satisfaction. Such research results highlight and indicate the significance of ecology for tourist destinations. The respondents expressed dissatisfaction with the damaged ecosystem, which directly affected their attitudes. Sociocultural and institutional dimensions of sustainability were singled out as the most significant by the respondents.

Stojanović et al. [8] point out that the involvement of residents in planning, developing, and controlling is extremely important for sustainable tourism within protected areas.

The promotion of sociocultural values can significantly improve tourism offerings. The realization of ecological, economic, and sociocultural benefits stands out as a significant pillar of sustainable tourism.

Primary action measures within tourism planning in protected areas include flora and fauna protection, limitation of natural resource exploitation [22], carrying capacity, zoning [24], implementation of ethical codes, strengthening the role of residents in tourism planning, development, and control [25], strengthening interaction between residents and visitors [21], and promotion of local products, events, heritage, construction of infrastructure without harmful effects on the destination environment and others [19,22,26–31].

Trišić et al. [32] examined the single impact of four sustainability dimensions on sustainable tourism development (hypotheses H1, H2, H3, and H4) within a protected area, i.e., the impact of sustainable tourism on the satisfaction of residents and visitors in a selected protected area. The sample consisted of 1419 respondents (789 residents and 630 visitors). The PoS model (prism of sustainability) was used in the research. The research results indicate that the ecological and sociocultural sustainability dimensions have the most considerable impact on sustainable tourism. The respondents rated the institutional dimension of sustainability (H1) the lowest, so this hypothesis was refuted. Concluding considerations include a series of planning measures to strengthen economic and institutional sustainability dimensions. In addition to residents' inclusion in tourism planning and development, it is crucial to promote nature protection, local culture, traditions, customs, cultural and historical heritage, and local craftsmanship. The forms of tourism that need to be developed within protected areas are nature-based tourism and ecotourism [33].

Research by Huayhuac et al. [34], included an examination of the impact of sustainable tourism on the local population of a protected area in Germany. Empirical research is based on the application of a survey questionnaire in collecting responses from residents. The PoS model (prism of sustainability) was used in the research. The survey model is adapted to examine and measure residents' perceptions of four sustainability dimensions: environmental, economic, sociocultural, and institutional.

The impact of sustainable tourism on residents' satisfaction was examined. The results of the research indicate that environmental sustainability is rated the most important. In addition to sociocultural and economic factors, sustainability brings the respondents the most significant satisfaction. The scientific contribution of this research is to provide valuable information about sustainable tourism development in protected areas. Furthermore, in this paper, the results have contributed to essential guidelines for examinations of sustainable tourism in the selected IPAs of Vojvodina.

Research by Cottrell et al. [35] aimed to examine the impact of sustainable tourism on residents living around two protected areas in Germany. The research methodology was also conceived according to the PoS model, i.e., in examining ecological, economic, sociocultural, and institutional sustainability to residents' satisfaction. Two research hypotheses were defined in the research:

**H1** —Four dimensions of sustainability prism (ecological, economic, sociocultural, institutional) are significant in the protected area.

**H2** —All four dimensions of sustainability have an impact on residents' satisfaction with sustainable tourism.

The research results confirm both research hypotheses. Respondents rated sociocultural and economic sustainability the most important. The obtained results helped the authors in designing the research model in this research.

In recent decades, tourism has had increasing impacts on the environment [4]. These impacts affect air, water, soil, and people. In numerous studies, the ecological, sociocultural, and economic influences of tourism have been singled out as the most significant. Some

of them leave lasting consequences on the environment [36]. From the abovementioned, protecting certain areas is a priority of many countries and managers of protected areas around the world. In protected areas and other destinations, it is necessary to develop tourism, but only those forms that will not have negative consequences [37–39].

Jeelani et al. [40] examined the attitudes of the local population in a protected mountainous area. A total of 352 households were interviewed using the survey technique and questionnaire as an instrument. Statistical data were processed with the help of the SUS-TAS method, which ranks and mutually rotates a total of 42 items positioned in 7 groups. The second objective of the research was to determine which dimension of sustainability has priority in the activities implemented in the protected area. An extremely high rating given to ecological sustainability indicates the developed awareness of the local population about the need and importance of protecting the space and living world from various influences to which these protected areas are exposed daily. The results of the research can be used to design strategies for the development of sustainable tourism in numerous protected areas.

Sustainable tourism development in protected areas can be examined through ecological, economic, sociocultural, and institutional sustainability [34,41]. A positive environmental dimension is realized by excluding the impact on the environment [42,43]. Economic sustainability is achieved through visitors' purchasing local products and services, employing residents, and strengthening the local economy [44]. Money earned from tourism activities directly increases employment, strengthening local products and services, and the role of the local population in planning and controlling tourism development [1,34]. Sociocultural sustainability is achieved through acceptable tourism impacts on the attitudes and satisfaction of residents and tourists with the state of tourism within protected areas as tourist destinations and through their mutual positive interaction [10,45,46]. Natural and social factors of protected areas strongly influence sustainable tourism [47]. Numerous specific forms of protected area tourism can also directly affect the economic, ecological, sociocultural, and institutional status of a destination with sustainable tourism [22,32,41].

It should be emphasized that sustainable tourism in protected areas can be examined with the aim of obtaining valid results by examining the four dimensions of sustainability. In addition to the economic, sociocultural and ecological dimensions, the idea of introducing the fourth dimension of sustainability first appeared in the study of the social implications of tourism development, which was created by Butler [48]. This research includes an overview of the impact of tourism development on different areas, with special attention focused on the social environment.

### 3. Study Areas

The area of international importance for plants—the IPA Obedska Bara—is located in the southeastern part of the province of Vojvodina. It extends over the territory of the SNR with the same name. The area is situated between 44°38′02″ and 44°46′05″ north latitude and from 19°47′16″ to 20°03′30″ east longitude. According to the method of creation, this protected area is a stagnant tributary, i.e., a cut meander of the Sava River, where the area of the pond and land is 25 km$^2$ [49]. Obedska Bara is one of the oldest protected areas in the world. The first protection dates back to 1874, and the last one from 1994, when it was declared an SNR with an area of 9820 ha and with a protection zone of 19,611 ha. Today, these wetlands occupy 12,000 ha, while the territory under protection has a total of 9880.43 ha [50]. This IPA is the largest wetland in Serbia. It is an ecosystem made up of bogs, ponds, swamp vegetation, wet meadows, and forests characterized by extraordinary opulence of biological diversity. There, the presence of rare and endangered plant and animal species of national and international importance is recorded. The Obedska Bara is especially important for the Sava River through the preservation and improvement of river water quality, for the spawning of fish, the reproduction of amphibians and reptiles, and the survival of rare and endangered species of aquatic plants. In 2005, this area was declared an important plant area (IPA), according to the decision of the PlantLife organization.

Furthermore, this SNR, Obedska Bara, has been designated as an area of Serbia that will be nominated for inclusion in the list of biosphere reserves within the UNESCO MAB—Man and Biosphere program [51].

More than 280 plant species have been recorded in the area of IPA Obedska Bara; many of them are rare, relict, endemic, and endangered, both in Serbia and in Europe. The area is dominated by various forest communities, in which *Quercus robur*, *Fraxinus angustifolia ssp oxycarpa*, *Carpinus betulus*, *Ulmus campestre*, *Acer campestre*, *Cornus mas*, *Acer tataricum*, etc., are strongly represented.

In the IPA Obedska Bara, the following habitat types of exceptional national importance are distinguished: white willow and poplar forests, areas of *Salix cinerea, Carex* spp., and communities of *Salvia natans*. Due to different biotic conditions, a complex and lush vegetation of forest, shrub, meadow, swamp, and water ecosystems is formed in this area. They represent ideal habitats for birds and many other animal species [49].

The IPA Zasavica is located in the southwestern part of Vojvodina. It extends between 44°52′56′′ and 44°58′04′′ north latitude and 19°24′07′′ and 19°36′31′′ east longitude. This area is protected as the SNR Zasavica, covering an area of 1128.55 ha, with a protection zone of 3462 ha [52]. The IPA covers a total area of 7923 ha. In addition, this protected area is also the Ramsar site in this part of Europe [2]. The entire reserve area along the river Zasavica is full of ponds and swamps that represent a unique spatial unit, significant for the protection of rare and endangered representatives of flora and fauna. According to the IUCN system of international protection, it is classified in the fourth category—an area of managing habitat or species, i.e., the area of the soil that is managed by active protection measures, in order to preserve habitats and ensure conditions for the survival of certain species [49]. In the IPA Zasavica, there are about 800 types of plants. The most significant endemically is *Aldrovanda vesiculosa*, for which Zasavica is the only habitat in Serbia. In addition to it, there are also *Stratiotes aloides, Trapa natans,* and *Aldrovanda vesiculosa*, which grow in the wetlands around the former Pannonian Sea [2,3]. The reeds and floating islands of Zasavica are inhabited by the relict species *Urtica kioviensis* and *Schaenoplectus triqueter*. A total of 22 non-native species of trees and shrubs originating from North America and Asia have been recorded in the reserve forests. Besides rare flora, the area of the reserve is inhabited by relict species of fauna, consisting of pond birds (IBA area), amphibians, fish (relict species *Umbra crameri*), insects and mammals (*Castor fiber, Lutra lutra, Myotis dasycneme, Myotis daubentonii*) [49,50].

The IPA and the SNR Koviljsko-Petrovaradinski Rit of the same name are located in Vojvodina, on the border between Bačka and Srem. The protected area extends between 45°11′34′′ north latitude and 20°02′10′′ east longitude [53]. It covers an area of 5895 ha in the territories of the municipalities of Novi Sad, Petrovaradin, Inđija, Sremski Karlovci, and Titel [2]. This is a complex of wetlands inhabited by extremely rare representatives of flora and fauna. The IPA status was established in 2005. This area is inhabited by: riparian forests of *Quercus robur, Fraxinus excelsior, Ulmus laevis, Populus nigra, Populus alba*, and *Salix alba,* then wet meadow, communities of *Dactylis glomerata, Numphar luteum, Nymphaea alba, Trapa natans*, etc. In this nature reserve, 443 taxa of higher plants have been recorded. *Salvinia natans, Callitriche palustris, Numphar luteum, Nymphaea alba, Trapa natans agg*, etc., stand out among the most important representatives of the flora. In addition to the rare flora, the area of the reserve is inhabited by significant representatives of the fauna: birds, fish, amphibians, reptiles, mammals, and insects [50]. The location of these three selected IPAs can be seen in Figure 1.

The current state of sustainable tourism in selected protected areas is not at a satisfactory level. Certain specific forms of tourism are developing in protected areas, such as short visits and sports tourism. Scientific research, ecotourism, and bird- and animal-watching tourism are developed at an initial stage. This research can contribute to the analysis of all potentials. The observed possibilities can indicate the strength of the forms of tourism that should be developed in selected areas, in order to ensure sustainability and to integrate protected areas into the tourist offer.

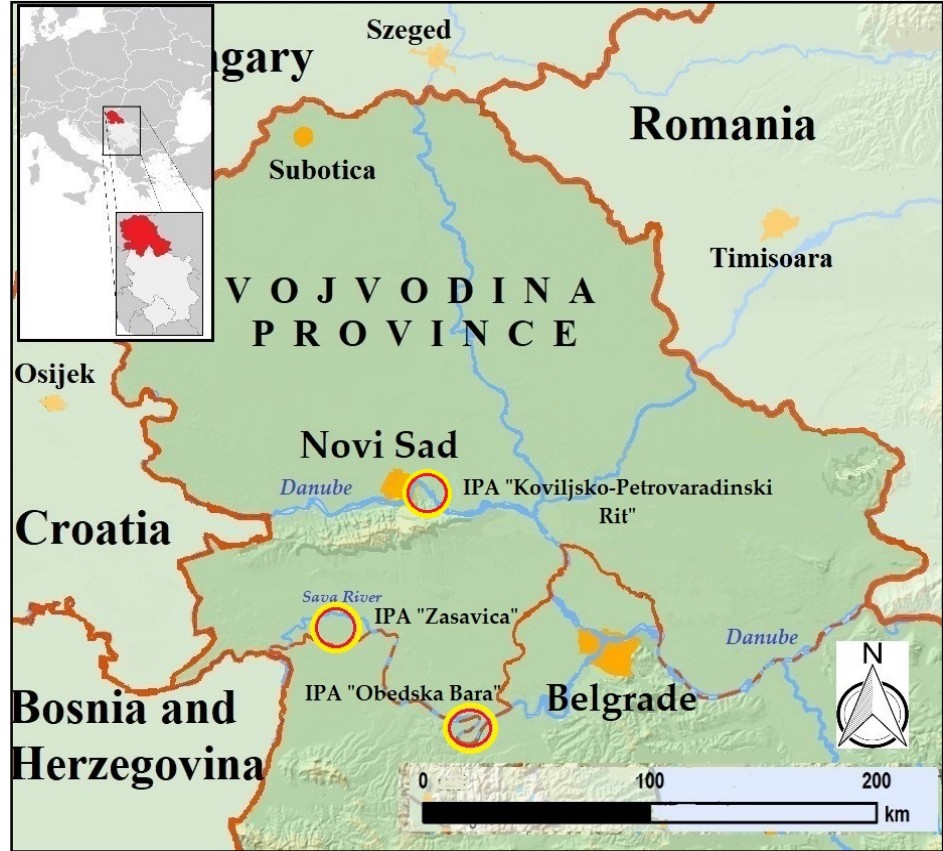

**Figure 1.** Map of important plant areas. Source: digitized by the authors.

The research area has national and international protection regimes. All three areas have national status—Special Nature Reserve. Within these areas, anthropogenic activities are limited. The areas are divided into I, II and III protection zones. In zone I, only habitat monitoring and scientific activities limited to observation and study of flora and fauna are allowed. Tourist activities are allowed in zone III. Plant and animal species inhabiting these protected areas are protected by national legal acts. In addition to national levels of protection, areas are also protected by international acts and statuses. The Ramsar Convention, IBAs, IPAs, PBAs and NATURA 2000 should be singled out as among the most significant [49,50].

Among the main indicators of tourism development in selected protected areas, the following can be distinguished: protection of endangered species, national and international protection regimes, development of tourist infrastructure, intensifying the role of residents, promotion of protected areas, development of nature-based tourism, cultural and scientific research tourism, economic benefits for residents and improving the financing of tourism development in protected areas [2,49–52]. Potential forms of tourism in selected protected areas can use the above indicators as potential values and a basis for the development of different forms of tourism.

## 4. Methods

This is a continuation of the research conducted by the same authors. By examining sustainable tourism in the protected areas of Vojvodina, the authors plan to complete the examination of sustainable tourism in protected areas into one territorial, geographical, and thematic scientific unit. This entirety is made up of selected protected areas of the Autonomous Province of Vojvodina (northern Serbia), in which there are tourist activities at a certain level of development.

There is a certain level of statistical error in the research and analysis of the obtained results. If we look at the size of the sample in relation to the total number of inhabitants in the settlements where the survey was conducted, it can be stated that with a confidence level of 95%, there is a possibility of a statistical error of $\pm 5\%$. When analyzing the collected responses from all questionnaires, it is concluded that all questionnaires are valid for analysis. The procedure for determining the validity of the questionnaire for analysis was performed personally and is an integral part of data collection in research and data preparation for statistical analysis.

### 4.1. Study Sample

The research PoS model was used in the paper based on research models according to Cottrell et al. [35], Huayhuaca et al. [34], and Trišić et al. [32]. The PoS model was adapted to the examination of sustainable tourism in three selected IPAs in Vojvodina. The conceptual model can be seen in Figure 2.

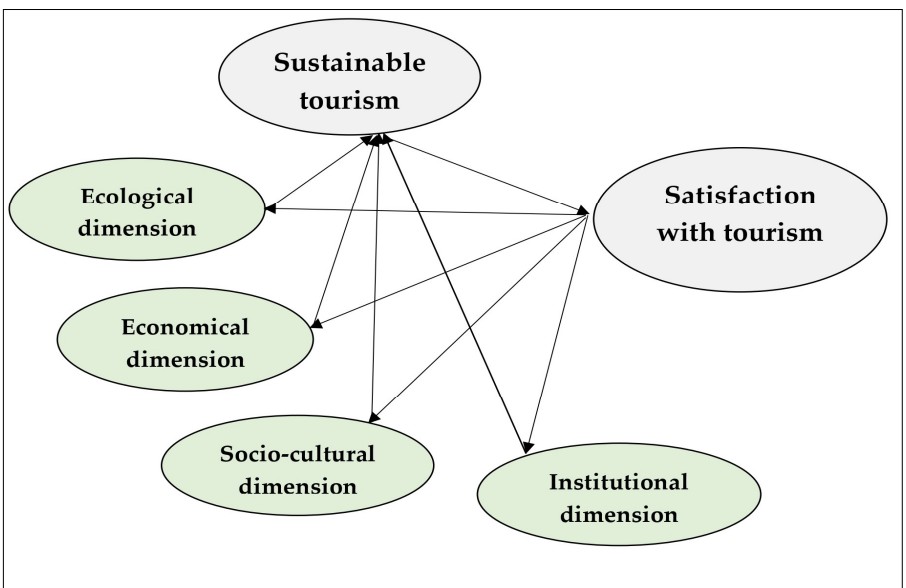

**Figure 2.** The conceptual research (PoS) model.

### 4.2. Instrument

The research is based on quantitative methodology, using the technique of interviewing respondents. The survey instrument is a questionnaire. In addition to the sociodemographic characteristics of the respondents, the questionnaire contains a total of 21 statements. The first 17 statements in the questionnaire are grouped into four dimensions of sustainability: the institutional, ecological, economic, and sociocultural dimensions of sustainability [54,55]. In addition to these statements, the questionnaire contains four statements directly related to residents' satisfaction with sustainable tourism development (Tables 1 and 2). Filling in the questionnaire is anonymous and does not contain data that could affect the identity of a respondent. Responding to the questionnaire, they grant their consent that the results of the survey can be used for scientific research purposes to examine sustainable tourism in protected areas and its impact on residents' satisfaction. Residents expressed their views using a five-point Likert scale (1—absolute disagreement, 5—absolute agreement, while grade 3 represents a neutral attitude) [44,56–62]. The research was carried out in the field and with the help of social networks. An online questionnaire was placed on social networks and user groups. A written questionnaire was filled in during personal contact with residents.

**Table 1.** Respondents' perceptions of the dimensions of sustainable tourism (*n* = 1134).

| Items | IPA "Zasavica" (*n* = 431) | | IPA "Obedska Bara" (*n* = 411) | | IPA "Koviljsko-Petro-Varadinski Rit" (*n* = 292) | |
|---|---|---|---|---|---|---|
| **Dimensions of Sustainable Tourism** | *α* | **Mean** | *α* | **Mean** | *α* | **Mean** |
| Institutional dimension | 0.796 | 3.30 | 0.803 | 3.14 | 0.769 | 3.43 |
| Visitors are guided through the protected area by trained guides and representatives of the local community | | 3.01 | | 2.86 | | 3.41 |
| Visitors in the protected area can see the local brands (wineries, ethno-houses, handicrafts, local enterprises, etc.) | | 4.02 | | 3.15 | | 3.02 |
| In the protected area, the manager's instructions on nature protection and visitor activities are followed | | 2.96 | | 3.11 | | 3.44 |
| Visitors are provided with information that reflects the history of the reserve, population and settlements | | 3.21 | | 3.43 | | 3.85 |
| Ecological dimension | 0.811 | 3.89 | 0.844 | 3.61 | 0.893 | 3.78 |
| There are roles of residents in protecting the area | | 3.41 | | 3.17 | | 3.52 |
| There are facilities, services and activities available to residents in the protected area | | 4.11 | | 4.25 | | 3.69 |
| There are tourist facilities without impacts on the environment | | 4.17 | | 3.41 | | 4.13 |
| Economical dimension | 0.769 | 3.19 | 0.753 | 2.98 | 0.740 | 2.94 |
| Protected area tourism creates economic benefits for residents | | 3.19 | | 3.05 | | 3.11 |
| Protected area tourism boosts the local economy | | 2.75 | | 2.62 | | 2.32 |
| Protected area tourism contributes to the residents' employment | | 2.88 | | 2.12 | | 3.09 |
| Local products are available to visitors | | 4.13 | | 3.88 | | 3.02 |
| Visitors support the prices of local products, tickets, etc. | | 3.02 | | 3.22 | | 3.17 |
| Sociocultural dimension | 0.801 | 3.75 | 0.864 | 3.44 | 0.812 | 3.31 |
| Visitors are interested in home products and crafts | | 4.51 | | 3.16 | | 3.09 |
| Visitors often and gladly come into contact with residents | | 4.02 | | 4.11 | | 4.00 |
| Visitors are interested in local traditions and customs | | 4.09 | | 3.95 | | 3.77 |
| Visitors visit local cultural facilities and events | | 3.02 | | 2.88 | | 2.77 |
| Visitors are interested in historical sites | | 3.11 | | 3.12 | | 2.96 |

Items measured on a 5-point Likert agreement scale. *α*—Cronbach alpha reliability.

**Table 2.** Scale items for the satisfaction index (*n* = 1134).

| Index | IPA Zasavica (*n* = 431) | | IPA Obedska Bara (*n* = 411) | | IPA Koviljsko-Petrovaradinski Rit (*n* = 292) | |
|---|---|---|---|---|---|---|
| | *α* | Mean | *α* | Mean | *α* | Mean |
| | **0.741** | **3.81** | **0.713** | **3.98** | **0.722** | **4.09** |
| I am satisfied that tourism in the protected area produces benefits for me and my family | | 3.14 | | 3.02 | | 3.09 |
| For me, it is important that there is sustainable tourism in this protected area | | 4.01 | | 4.26 | | 4.44 |
| I am satisfied because tourism contributes to increasing the attractiveness of this protected area | | 4.66 | | 4.54 | | 4.71 |
| I am satisfied with the state of tourism in this protected area | | 3.42 | | 4.11 | | 4.14 |

Using an adapted PoS research model in examining sustainable tourism in protected areas can help define the state of sustainable tourism [32]. Additionally, it can indicate potential weaknesses and threats to tourism development. Surveying residents with direct connections to the protected area can provide appropriate results regarding the possibility of developing certain forms of tourism within these areas. Those forms of tourism can contribute to individual ecological, economic, institutional, and sociocultural dimensions of sustainability [55].

*4.3. Procedure*

The survey was conducted from March to October 2022. For research purposes, we selected the respondents by random sampling. In the protected area of Obedska Bara, the residents were surveyed in the following settlements: Kupinovo, Obrež, Grabovci, Ašanja, and Ogar (411 residents in total). The survey in the IPA Zasavica was carried out in the populated areas: Crna Bara, Banovo Polje, Ravnje, Radenković, Zasavica, Salaš Noćajski, Noćaj and Mačvanska Mitrovica (a total of 431 residents). The study of sustainable tourism in the IPA Koviljsko-Petrovaradinski Rit was carried out in populated areas: Kovilj, Sremski Karlovci, Gardinovci, Beška, Čortanovci and Krčedin (a total of 292 respondents). The settlements where the research was carried out are located around the protected zones of the selected protected areas. As for the distance of settlements from the protection zone of the Special Nature Reserve Obedska Bara, it can be stated that the settlements of Kupinovo, Grabovci and Obrež are on the very edge of the reserve. The settlement of Ogar is about 7 km from the nearest line of the protection zone, while the settlement of Ašanja is about 3 km from the border of the reserve. Regarding the location of the settlements around the protected area Zasavica, the nearest settlement is Zasavica and it is located next to the reserve itself. The most distant settlement where the survey was conducted is Mačvanska Mitrovica, which is slightly more than 5 km from the border of the reserve. As for the settlements where sustainable tourism was tested in the protected area of Koviljsko-Petrovaradinski Rit, it can be stated that all the settlements are located in the immediate vicinity of the protection zone of the reserve, i.e., at a distance of about 1 km. The population that participated in the survey, as well as the places where the surveyed residents live, are in direct geographical, economic, or social relation to the research areas. Respondents were selected using random sampling. During personal contact, they were given written questionnaires to fill out. The survey was completely anonymous. By filling out the questionnaire, the respondents gave their consent that the results of the research can be used for scientific purposes. One part of the questionnaire, about 60%, was filled out with the help of social networks and thematic groups. The validity of the given answers is ensured by detailed instructions before filling out the questionnaire. Respondents could

opt out of filling out the questionnaire at any time. The residents who participated in the survey have direct connections with the protected areas that are the subject of analysis in this paper.

*4.4. Statistical Analysis*

The authors analyzed and presented the collected data using the Statistical Package for Social Science (SPSS.21) software. We applied Cronbach's alpha analysis to test and measure the reliability of the obtained answers connected with the four dimensions of sustainability and the degree of residents' satisfaction with sustainable tourism, i.e., to test the reliability of dependent and independent variables [57]. We also used regression analysis in order to examine the degree of residents' satisfaction with sustainable tourism [10,35]. The obtained results were subjected to a comparative analysis for all three protected areas, and are presented in a tabular manner. The level of reliability of the samples will be examined with the help of the Cronbach alpha coefficient. The validity of the obtained answers of all four dimensions of sustainability will be examined with the help of Cronbach's alpha. The level of sampling error will be reduced as the sample size increases. Error testing will be done using *T*-test analysis, where the standard deviation of the set is not known.

Using an adapted PoS research model in examining sustainable tourism in protected areas can help define the state of sustainable tourism [32]. It can also indicate potential weaknesses and threats to tourism development. Surveying residents with direct connections to the protected area can provide appropriate results regarding the possibility of developing certain forms of tourism within these areas. Those forms of tourism can contribute to individual ecological, economic, institutional, and sociocultural dimensions of sustainability [55].

Studying the past of area protection and the role of residents in the observed areas, the following can be stated. The oldest protection status in these areas is the protected area Obedska Bara. The status of protection dates back to 1874. The spaces of all three protected areas have always been inhabited by rare species of plants and animals. Unnecessary exploitation of resources, poaching, harmful effects of chemical agents from the surrounding agricultural lands and improperly developed tourism have caused certain species to be on the red list of threats, and in need of additional protection. The representatives of the local population had to make only selective use of resources from the protected areas, which exclusively serves for their personal existence. The expansion of tourist activities in reserves is linked to the 20th century, when more frequent group visits to these protected areas began. This also affected the development of the tourist infrastructure, which in some parts of the protected areas was not in accordance with ecological principles and the overall environment. This caused the state to enact a number of new regulations that are exclusively related to nature protection and the control of tourist development. Protection zones have been established in the nature reserves, in which case tourism is allowed in zone III. Scientific research activities are carried out in zone I. In the past, residents had a negligible role in the protection of the area and the development of tourism. Today, their role is more significant because residents are involved in tourism development planning, management processes, and in the control of protection and tourism development.

## 5. Results

A total of 1134 residents were surveyed. All survey questionnaires were valid and subjected to statistical analysis. Of the total number of respondents, 56% are women. The average age is 36 (from 18 to 81). The respondents' educational structure is as follows: 66% of respondents have secondary education, 19% of respondents have primary education, 14% of respondents have college or higher education and a total of 1% of respondents have master's and PhD degrees.

The statistical processing of data included an examination of the variables' reliability in order to examine all four dimensions of sustainability and the impact of sustainable tourism

on residents' satisfaction. Each dimension of sustainability was calculated separately, which resulted in gaining the average value of the variables [32–35] (Table 1).

The average values for the four dimensions of sustainability for each IPA can be seen in Figure 3.

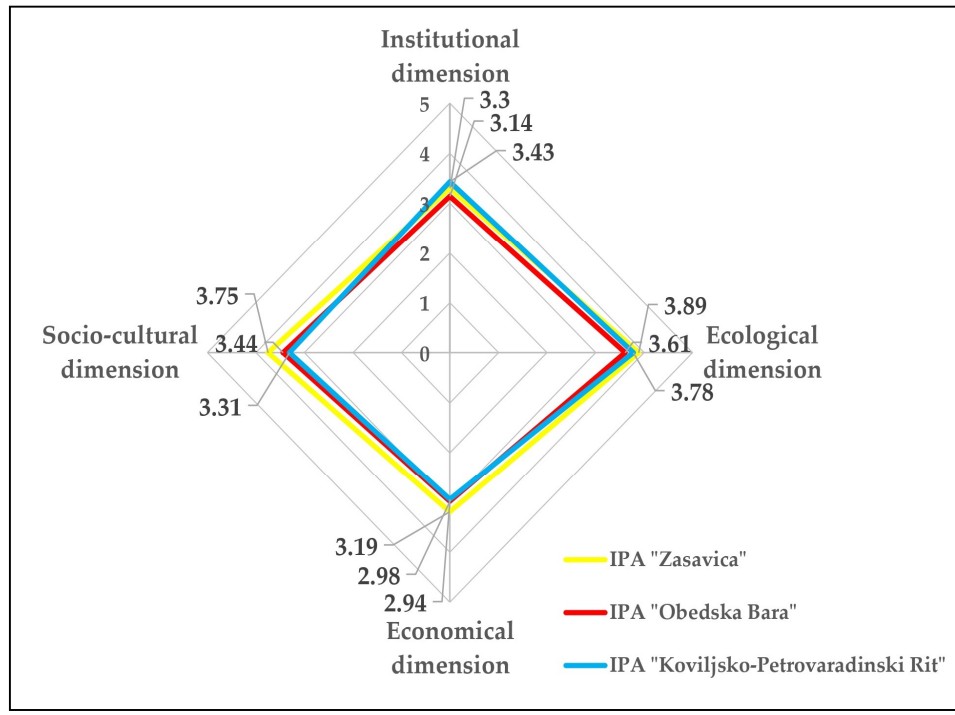

**Figure 3.** Presentation of sustainability dimensions average values for selected IPAs.

The overall mean satisfaction with sustainable tourism development in the observed protected areas can be seen in Table 2.

By applying regression analysis, it can be determined whether each dimension of sustainability contributes to residents' satisfaction with sustainable tourism development [9,10,35]. The assumption includes the results of four dimensions of sustainability, and the results of satisfaction with sustainable tourism, accounting for 29.3% (the IPA Zasavica), 28.1% (the IPA Obedska Bara), and 31.1% (the IPA Koviljsko-Petrovaradinki Rit) of the variances, were explained ($R_1^2 = 0.293$, $R_2^2 = 0.281$, and $R_3^2 = 0.311$) (Table 3).

**Table 3.** Regression analysis of each protected area on resident satisfaction ($n = 1134$).

| Satisfaction with Tourism | IPA Zasavica ($n = 431$) | | IPA Obedska Bara ($n = 411$) | | IPA Koviljsko-Petrovaradinski Rit ($n = 292$) | |
|---|---|---|---|---|---|---|
| | $\beta_1$ | *p*-Value | $\beta_1$ | *p*-Value | $\beta_1$ | *p*-Value |
| Institutional dimension | 0.356 | 0.000 | 0.396 | 0.000 | 0.339 | 0.000 |
| Ecological dimension | 0.411 | 0.000 | 0.429 | 0.000 | 0.401 | 0.000 |
| Economic dimension | 0.489 | 0.000 | 0.491 | 0.000 | 0.423 | 0.000 |
| Sociocultural dimension | 0.511 | 0.000 | 0.533 | 0.000 | 0.499 | 0.000 |

Standardized $\beta$ value used. $R_1^2 = 0.293$, $R_2^2 = 0.281$, $R_3^2 = 0.311$.

## 6. Discussion

If we analyze the obtained average values of individual dimensions of sustainability, the concluding considerations indicate that two dimensions of sustainability have more significant values. These are the sociocultural and ecological sustainability dimensions. The average values of the sociocultural dimension are 3.75, 3.44, and 3.31, and of the

ecological dimension 3.89, 3.61, and 3.78. Sociocultural and ecological dimensions of sustainability achieve the greatest impact on sustainable tourism in the IPA Zasavica. Factors that indicate nature protection, the interaction between residents and visitors, and the promotion of local products and culture, are more expressed in this protected area. The ecological dimension has the lowest average value in the IPA Obedska Bara. However, the sociocultural dimension has the weakest impact on sustainable tourism in the IPA Koviljsko-Petrovaradinski Rit. This information may indicate that it is necessary to significantly develop those forms of tourism based on nature and the promotion of cultural values. The economic dimension of sustainability (3.19, 2.98, and 2.94) and the institutional dimension of sustainability (3.30, 3.14, and 3.43) have slightly lower average values of the variables. Sociocultural and ecological dimensions are singled out as the most significant dimensions of sustainability due to the natural and social factors that characterize these areas. Within them, there are different and very rare animal and plant species. Their survival is very important for residents and the entire population. Emphasizing the importance of ecological factors indicates the growth of environmental awareness among the population, which implies that it is possible to develop nature-based forms of tourism in protected areas. The population living next to protected areas has developed cultural potential, tradition and heritage. This was recognized by the respondents as an important tourist opportunity that needs to be properly built up. When creating tourism development strategies for these protected areas, the ecological and sociocultural dimensions of sustainability should be separately analyzed and implemented in tourism plans [7–10]

The economic dimension achieves the highest impact on sustainable tourism in the IPA Zasavica, while the institutional dimension has the highest value in the IPA Koviljsko-Petrovaradinski Rit. From the analysis of all the values of individual sustainability dimensions, it can be concluded that the four dimensions of sustainability have the most significant impact on sustainable tourism in the IPA Zasavica, while the lowest one is in the IPA Koviljsko-Petrovarinski Rit (Table 1 and Figure 3). Reliability testing of the variables indicates that $\alpha > 0.65$ for all dimensions of sustainability, and that reliability can be accepted [35,63] in all protected areas.

The results of sustainability dimension measurement analysis may indicate the existence of certain strengths, opportunities, and weaknesses in tourism planning and development. When we create tourism planning strategies, these data can be crucial. It is necessary to develop those tourism forms that have the task of improving weaker or average values of sustainability dimensions. Potential models in all three IPAs are nature-based, scientific research, events, gastronomic and cultural, trips, sports and recreational, and other tourism forms. In planning and monitoring tourism development, it is crucial to include residents and intensify their interaction with visitors [64].

By analyzing the data in Tables 2 and 3, after applying the regression analysis, it can be concluded that residents are satisfied with sustainable tourism development. Contrary to the values of the individual sustainability dimensions, satisfaction with sustainable tourism is most explicit among the residents of the IPA Koviljsko-Petrovaradinski Rit. Considering that the institutional sustainability dimension has the highest value in this protected area, it can be related to the organization of tourism and management activities. This indicates that management processes are somewhat more expressed in this reserve. The implications are the functioning of visitor centers, guides, and expert services, as well as adherence to various guidelines and instructions issued by governing institutions. By analyzing the individual values of sustainability dimensions and residents' satisfaction with sustainable tourism, it can be concluded that the proper development in the observed areas can directly affect individual tourism values of institutional, ecological, economic, and sociocultural sustainability dimensions, and also the greater satisfaction of residents with the sustainable tourism development.

If we specifically analyze the impact of sustainable tourism on the satisfaction of residents (Table 3), it can be concluded that the standardized β value tends to increase and strengthen the value. This indicates the existence of an influence of the four dimensions

of sustainability on the degree of satisfaction of residents with sustainable tourism. By analyzing the data presented, it can be concluded that the impacts of the dimensions of sustainability have relatively identical values. Which of the four dimensions of sustainability has the greatest impact on the respondents and which individual factors have the most significant values will be the subject of future research by the authors.

In the authors' previous research, the sustainable tourism of special nature reserves in other parts of Vojvodina was examined. The selected areas differ in their structure in terms of natural and social factors. These are favorable geographical position, specific landforms, the existence of wetlands or another hydrographic potential, rare endemic or threatened flora and fauna, significant national and international protection statuses of the area, tourist infrastructure and the possibility of its development, tourist traffic, capacities for tourist accommodation, and the possibility of development of specific forms of tourism. The authors plan to examine the state of sustainable tourism in as many protected areas as possible in a geographical and territorial entity, such as Vojvodina. Therefore, this paper is a continuation of research on sustainable tourism.

In previous research, the authors used the PoS model to examine the state of sustainable tourism through four dimensions of sustainability: institutional, ecological, economic and sociocultural dimensions. In addition, the research included measuring the impact of sustainable tourism on the satisfaction of residents and visitors of protected areas. In a large number of protected areas, respondents singled out the ecological and sociocultural dimensions of sustainability as the most significant and the dimensions with the greatest impact on sustainable tourism. So far (the authors) have examined nearly 10,000 respondents using the survey technique, with the help of a questionnaire as an instrument. The sample consisted of residents, visitors and experts for protected areas. As the selected areas differ in their structure and potential for tourism development, the obtained results coincide with the results of the research in this paper. Environmental and sociocultural factors have been singled out as the most important for the development of sustainable forms of tourism. This indicates the constant need to protect the area and the developed awareness of the importance of nature protection, because only non-degraded areas can be real sustainable tourist destinations. This information indicates that when planning and developing tourism, it is necessary to develop nature-based forms of tourism. The promotion of the ecological values of these protected areas is a significant management activity. In this and previous research by the authors, the sociocultural dimension of sustainability was recognized as important by the respondents, given that specific cultures, traditions and heritage are developing in these areas. The promotion of ethno-social values can be an important tourist activity, because it can contribute to the creation of new tourist products and an increase in importance on the tourist market. In earlier surveys, visitors expressed great interest in local products, local culture, facilities and local events. In all papers, we emphasize the importance of more intensive involvement of residents in the overall planning and development of tourism. The local population is a pillar of sustainable tourism. In addition to the above, the research results indicate the importance of developing more intensive interaction between residents and visitors in protected areas. This can be successfully realized precisely by involving residents in tourism development through various specific forms of tourism, and in particular, nature-based types of tourism.

When we compare the current results with the authors' previous research, a common parallel is evident, which singles out the protection of nature and the promotion of cultural values as activities that currently have the greatest impact on sustainable tourism. The institutional dimension of sustainability was identified in the previous and in this research with a slightly lower impact on sustainability. This indicates that it is necessary to significantly develop legislative and management processes, which coincides with research on sustainable tourism, conducted by other authors [7,8]. The economic dimension of sustainability was singled out as important, but with a lower impact on sustainable tourism. The reasons for this may be due to the weak development of tourism in these protected areas, which affects the lower consumption of tourists and the lower level of employment of

the local population. It is necessary to adopt a set of plans that will contribute to the growth of tourism and the greater economic benefits of tourism for residents. The proper evolution and massification of tourism can be one of the tools for the development of sustainable tourism. This coincides with research on sustainable tourism by domestic and foreign authors, who examined sustainable tourism in different protected areas of the world [7–10].

While researching sustainable tourism in the protected areas of Vojvodina, the authors aim to make the results reliable. That is why they are planning to examine sustainability in as many different protected areas as possible in this part of Serbia. When the results of the research are compared with similar results from the environment, Europe and the world, a relative match can be established. The results of research on sustainable tourism by Stojanović et al. [8] indicate that for sustainable tourism in protected areas, the inclusion of residents in the planning system, development and control of tourism development is extremely important, and ecological and sociocultural sustainability were evaluated by the respondents as the highest values. This coincides with the research in this article. Achieving ecological, economic and sociocultural benefits for residents, visitors and the protected area stands out as an important pillar of sustainable tourism. The results of the research of Asmelash and Kumar [23] indicate the importance of ecological principles within protected areas. Residents and visitors expressed dissatisfaction with the damaged ecosystem. Sociocultural and institutional dimensions of sustainability were singled out as the most significant by the respondents. Using the results of sustainable tourism research using the PoS model, Huayhuaca et al. [34] indicate that environmental sustainability is the one with the greatest impact on sustainable tourism. In addition to sociocultural and economic, this sustainability brings the most significant satisfaction to the respondents. The results of the research have important similarities with the results of the research in this paper, where ecological sustainability is also singled out as a meaningful dimension of sustainability by the respondents. In a study by Cottrell et al. [35], the PoS model was also applied. The research was designed to examine the impact of sustainable tourism on residents living around two protected areas in Germany. The results of the research match the results of the research in this paper. Respondents rated sociocultural and economic sustainability as the most important dimensions. The institutional dimension of sustainability is singled out as important, which represents a difference in relation to the research results in this paper. If we combine the results of the research in this paper with the results of research on sustainable tourism in protected areas in the world, it can be concluded that it is necessary to significantly develop forms of tourism that will aim to strengthen institutions and that will contribute to important economic benefits for residents. Residents should be more involved in the creation of legal regulations concerning the protection of the area and in studies and planning of tourism development. It is also necessary to develop a better interaction between residents and visitors. The promotion of ethno-social values in the development of tourism in protected areas should be very important. Special attention must be paid to the greater participation of the local population in education and tourist guiding activities. This is very important because of the training of local people to provide services to tourists.

## 7. Conclusions

Within the tourism sector, the importance of ecologically sustainable development is constantly growing, which is a condition for balancing the care for nature and at the same time the development of tourism. The problems of protecting the natural environment are constantly increasing, so measures must be taken urgently to prevent and rehabilitate the degradation of nature.

Analyzing the natural and social potentials and values of the selected IPAs, we can conclude that these protected areas show the potential to improve nature on one side and to develop some specific tourism forms on the other. In the examined protected areas, there are exceedingly rare flora, characteristic only for these areas. Additionally, the basic ecosystem consists of wetlands inhabited by representatives of fauna. Due to the abovementioned

rarity and prevention of anthropogenic influences, these areas are protected as special nature reserves. In addition, international protection regimes have been established.

Certain tourist activities are carried out in the examined IPAs. By all means, some activities have an impact on the environment, too. The absence of measures and control of tourism development can have significant negative tourism impacts on the environment [65–67]. If the proper development of tourism ensures ecological, economic, sociocultural, and institutional improvements of these destinations' value, we can talk about a certain level of sustainable tourism [35,68–70]. Providing benefits for the protected area and residents, either material, social, or ecological, is a basic prerequisite for the constitution of sustainable tourism forms [71,72]. Considering that the selected IPAs are fundamentally sensitive ecosystems, it is necessary to develop nature-based tourism forms. One of the models would be ecotourism and scientific research tourism forms. This implies controlled construction of tourist infrastructure without harmful effects on the environment [73], the inclusion of residents in tourism planning, development and control, education of visitors, application of ethical codes, space zoning, and use of carrying capacity. Strengthening ecological concepts and a healthy environment has stimulated governments of numerous countries and managers of protected areas to take responsibility in such a way that there is an expansion of tourism in protected areas. The motivation of visitors to go to nature is an essential element when studying the types of tourist demand and finding segments crucial for visiting a preserved natural environment.

Contemporary tourism trends and the business environment are extremely changeable and innovative. Tourism development in protected areas can also result in significant economic benefits. Scientific and research tourism, ecotourism, excursion tourism, bird- and animal-watching tourism, safari tourism, events, and cultural tourism could provide financial resources for education, research, and protection of areas and species [27,74,75]. Ecotourism can benefit residents in an ecological, cultural, and economic sense. Unlike other participants in tourism movements, an ecotourist enjoys nature with a local guide, stays in local accommodations, uses local products, and supports local culture and economy.

The promotion of tourism development in the protected area is a priority in all planning and development activities [76]. It can be a means of reaching users and consumers of tourism services [77]. The results of the research conducted in this paper indicate that the promotion of selected protected areas should be based on the ecological and sociocultural values of these tourist destinations. Today, nature protection is becoming a primary activity in planning tourism development all around the world. In addition to the abovementioned, it is necessary to promote social values that can influence to create a direct interaction between visitors and residents [78–80]. Tourism focused on destinations in nature is essentially a trip to areas with beautiful and unspoiled nature.

The originality of this research is reflected in the fact that in these protected areas, for the first time, the impact of sustainable tourism on the satisfaction of residents is examined, using the PoS research model. So far, tourism and the factors that can influence the development of tourism have been investigated. In the observed territory of the protected areas, research using the four dimensions of sustainability have not been applied until now. This was done in this research. As the ecological and sociocultural dimensions of sustainability have been identified as the most important dimensions of sustainability, planning measures for the protection and development of tourism must rely on natural and social potential.

### 7.1. Practical Implications

The concluding considerations indicate that the promotion of IPAs' natural and social values represents a primary activity of tourism development. The development of sustainable forms of tourism in protected areas can represent a model for the reduction or elimination of negative tourism impacts [81]. Additionally, tourism can influence the creation of a large number of positive effects on the living and social environment. With the application and control of certain measures of nature protection and tourism development,

visitors and residents create their own perceptions related to the ecological, economic, sociocultural, and institutional benefits of sustainable tourism. Satisfaction with sustainable tourism is precisely the important result of sustainable tourism development planning.

In the selected IPAs, we examined sustainable tourism, its condition, and its impact on residents' satisfaction. By expressing their perceived attitudes, residents evaluated sustainable tourism through four dimensions of sustainability: institutional, ecological, economic, and sociocultural. Each of the mentioned dimensions exerts certain influences on the overall state of sustainable tourism. Ecological and sociocultural dimensions were assessed as the dimensions with the greatest impact on sustainable tourism in all three observed protected areas. Considering the obtained average values for all three observed protected areas, we can conclude that the values are relatively identical. This indicates that the tourism development strategies in this part of Vojvodina can have mutual and unique planning measures. The extent to which each of the measures would achieve results individually will be the subject of future authors' research. They will also examine the attitudes and satisfaction of visitors with sustainable tourism in these protected areas. In order to create a unique picture and achieve more reliable scientific results, in future research, the authors will implement the data collection of managers' and experts' views on tourism development in these protected areas.

The implications of the scientific research results in this paper can be reflected in their influence and help in the development of different tourism development strategies, which include tourism of protected areas as specific tourist destinations. In the protected areas of Vojvodina, sustainable tourism has not been developed at a satisfactory level, like the sustainable tourism of protected areas in certain parts of the world. In addition to the frequent absence of control measures, there are also financing issues. The method of providing the necessary finances for the development of sustainable tourism forms within protected areas is an important segment of management processes.

### 7.2. Limitations and Future Recommendations

Constant limitations in research work are often found both in the absence of understanding at the local level and in the lack of financial resources. Since these studies are carried out in selected protected areas, establishing a connection with local administrations is the basis for successful work. Obstacles and misunderstandings are often encountered, which affect the extension of the study, and often lead to the suspension of work. For this reason, authors are often faced with a limitation of the possibility of using the resources needed for research. The authors of this paper, through constant and extensive research, managed to establish a pattern of trust with local self-governments and in this way convinced the local community that these researches are necessary for its further development.

The obtained research results can be used to examine sustainable tourism in other protected areas, not only in Vojvodina and Serbia but also in other parts of the world. Additionally, the results can provide important guidelines when writing tourism development strategies, protection studies, creating spatial plans, financial plans and other legal acts. The authors will expand their future research to other protected areas in Vojvodina (northern Serbia). By expanding the research area, a larger geographical and territorial entity will be included, so that the results of the research on the state of sustainable tourism will be more reliable. After that, the examination of sustainable tourism will spread to the protected areas of the region, in order to perform a detailed comparative analysis. This will enable the identification of weaknesses and opportunities for the development of sustainable forms of tourism. Ways of providing more financial investments in tourism of protected areas will also be the subject of the authors' future research.

**Author Contributions:** Conceptualization, I.T., D.M., V.R., F.N., M.M., S.Š. and A.N.C.; methodology, I.T., D.M., V.R., F.N., M.M., S.Š. and A.N.C.; software, I.T., F.N., M.M. and A.N.C.; validation, I.T., D.M., V.R., M.M., S.Š. and A.N.C.; formal analysis, I.T., D.M., V.R., F.N., M.M. and S.Š.; investigation, I.T., D.M., V.R., F.N., S.Š. and A.N.C.; resources, I.T., D.M., F.N., M.M., S.Š. and A.N.C.; data curation, I.T., D.M., V.R., F.N., M.M., S.Š. and A.N.C.; writing—original draft preparation, I.T., D.M., V.R., F.N.,

M.M., S.Š. and A.N.C.; writing—review and editing, I.T., D.M., V.R., F.N., M.M., S.Š. and A.N.C.; visualization, I.T., V.R., F.N., M.M., S.Š. and A.N.C.; supervision, I.T., D.M., V.R., F.N., M.M., S.Š. and A.N.C.; project administration, I.T., D.M., V.R., F.N., M.M. and A.N.C.; funding acquisition, I.T., D.M., V.R., F.N., M.M., S.Š. and A.N.C. All authors have read and agreed to the published version of the manuscript.

**Funding:** This research received no external funding.

**Data Availability Statement:** The data that support the findings of this study are available upon reasonable request from the authors.

**Conflicts of Interest:** The authors declare no conflict of interest.

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
