# Peer review of "Sustainable Tourism of Important Plant Areas (IPAs)—A Case of Three Protected Areas of Vojvodina Province"

_land, doi:10.3390/land12071278_

Round 1

Reviewer 1 Report

Dear Author(s),
Thank you for the opportunity to read the paper entitled Sustainable Tourism of Important Plant Areas (IPAs) – a case of 2 Three Protected Areas of Vojvodina Province.

I found this paper interesting. The topic of this paper is interesting but improvements are very much necessary.

Abstract

Comment 1

Lines 20-21 - You should modify the first sentence or insert a new one before it. This is not the way to start the abstract.

Comment 2

Lines 36-37 – The last sentence is unclear. You should modify it.

Introduction

Comment 3

What is the originality of the research? Contribution to the research field? You should point out previous research and the necessity of your work.

Comment 4

Lines 79-83 – This does not belong to the Introduction. It is part of the Methodology.

Study areas

Comment 5

Line 217 – “wet habitat” – Do you mean wetlands? English proofreading is necessary.

Comment 6

Lines 228-235 References?

Comment 7

Lines 245-267 – References just in end. Please cite properly and modify this.

Comment 8

Line 272 – missing “.”.

Methodology

Comment 9

Divide the section Methodology into subsections “Study sample” “Instrument” “Procedure” “Statistical analysis” etc.

Comment 10

It is necessary to better explain the study sample. How did you determine the sample? How did you collect the questionnaire? What was the margin of error, confidence level?

Comment 11

What is the current state of tourism and protection in these areas? This is unclear and missing.

Discussion

Comment 12

You need to make a proper discussion out of your results and compare it to previous studies.

Comment 13

377-388 – Why are they the most expressed in this area? What is your opinion based on the real situation in the SNR? Do you have some official data to compare it and explain this?

Comment 14

Same as the previous comment for SNR Obedska Bara and Koviljsko-Petrovaradinski Rit.

Comment 15

The discussion section needs serious rework. This is just an interpretation of results and repeating from tables.  It is necessary to compare and explain your results, to compare them with actual projects from these areas and from other cases in Europe and around the world. The scale you used is really a lot used in the literature, so it would be nice to make comparisons.

Conclusions

Comment 16

Focus on showing the originality of your work. Novelty? What are theoretical contributions? Management implications?  What are the limitations? You should make subsections like “Theoretical implications”, “Practical implications”, “Limitations and future recommendations” etc.  

Once again thank you very much for the opportunity to read this interesting article. The manuscript has really nice results, but improvements would be appreciated. Looking forward to reading your article again.

Wish you all the best!

Reviewer

Some sentences are very difficult to comprehend. Extensive English editing is required. 

Author Response

Dear reviewer,

we want to inform you that we have taken all your suggestions into account. According to each suggestion, we made corrections in the text. We are grateful for the suggestions because they contributed to the quality of this research. We inform you of the following:

Abstract

Comment 1

Lines 20-21 - You should modify the first sentence or insert a new one before it. This is not the way to start the abstract.

Answer: we made a correction in the work. Lines: 20-21 new.

Comment 2

Lines 36-37 – The last sentence is unclear. You should modify it.

Answer: corrections are marked with red text, lines: 38-39.

Introduction

Comment 3

What is the originality of the research? Contribution to the research field? You should point out previous research and the necessity of your work.

Answer: the supplement in the text is marked in red. Lines: 82-95.

Comment 4

Lines 79-83 – This does not belong to the Introduction. It is part of the Methodology.

Answer: the text has been deleted and moved to the Methodology chapter. Lines: 96-100.

Study areas

Comment 5

Line 217 – “wet habitat” – Do you mean wetlands? English proofreading is necessary.

Answer: the correction was made in the text. Line: 241.

Comment 6

Lines 228-235 References?

Comment 7

Lines 245-267 – References just in end. Please cite properly and modify this.

Comment 8

Line 272 – missing “.”.

Answer: We have corrected the references in the text. Lines: 264, 268, 269, 275.

Methodology

Comment 9

Divide the section Methodology into subsections “Study sample” “Instrument” “Procedure” “Statistical analysis” etc.

Answer: We have made all necessary corrections to this chapter. We have created sub-chapters according to your suggestions. Lines: 344 – 421.

Comment 10

It is necessary to better explain the study sample. How did you determine the sample? How did you collect the questionnaire? What was the margin of error, confidence level?

Answer: We have expanded the new chapters with additional explanations of how we did the sampling, how the respondents were chosen, how the potential error was controlled and how the greater reliability of the given answers and samples was ensured. Lines: 381-399; 409-414; 415-421; 490-493.

Comment 11

What is the current state of tourism and protection in these areas? This is unclear and missing.

Answer: We have expanded the Study Area chapter with the necessary explanations in everything according to your suggestions. Lines: 313-328.

Discussion

Comment 12

You need to make a proper discussion out of your results and compare it to previous studies.

Answer: We have expanded the Discussion chapter in everything according to your suggestion, for which we are especially grateful. Lines: 524-582.

Comment 13

377-388 – Why are they the most expressed in this area? What is your opinion based on the real situation in the SNR? Do you have some official data to compare it and explain this?

Comment 14

Same as the previous comment for SNR Obedska Bara and Koviljsko-Petrovaradinski Rit.

Answer: We have expanded the Discussion chapter for text that concerns the ecological and socio-cultural dimensions of sustainability. Lines: 474-485; 524-582.

Comment 15

The discussion section needs serious rework. This is just an interpretation of results and repeating from tables.  It is necessary to compare and explain your results, to compare them with actual projects from these areas and from other cases in Europe and around the world. The scale you used is really a lot used in the literature, so it would be nice to make comparisons.

Answer: We adapted the Discussion chapter in everything to your comments. Lines: 460-582.

Conclusions

Comment 16

Focus on showing the originality of your work. Novelty? What are theoretical contributions? Management implications?  What are the limitations? You should make subsections like “Theoretical implications”, “Practical implications”, “Limitations and future recommendations” etc.

Answer: We expanded the Conclusion chapter and divided it into subchapters. Lines: 583-701.

We thank you for your extraordinary trust,

Authors.

Reviewer 2 Report

Thanks for the paper. I suggest mentioning the country in the article's title or at least in the abstract. The overview of three case study areas provides readers with information on nature values, plants, and species, but there is a lack of giving an explanation on tourist flow and tourism infrastructure. The advice is to add a short review of the main indicators that illustrate tourism in case study areas.

The conceptual research model includes sustainable tourism, satisfaction with tourism and four sustainability dimensions. I suggest strengthening the explanation of dividing separately "satisfaction with tourism".  Provide a precise title of the satisfaction index, for example, Table 2.

Author Response

Respected Dear,

first of all, we want to thank you for your trust! After that, let us inform you that we have taken all your suggestions into account. According to each suggestion, we made corrections in the text. We are grateful for the suggestions because they contributed to the quality of this research.

We inform you of the following:

I suggest mentioning the country in the article's title or at least in the abstract. 

Answer: we have changed the text: Lines: 20-21.

The overview of three case study areas provides readers with information on nature values, plants, and species, but there is a lack of giving an explanation on tourist flow and tourism infrastructure. The advice is to add a short review of the main indicators that illustrate tourism in case study areas.

Answer: we expanded the text according to your suggestions. Lines: 312-336.

The conceptual research model includes sustainable tourism, satisfaction with tourism and four sustainability dimensions. I suggest strengthening the explanation of dividing separately "satisfaction with tourism". 

Answer: we added an explanation for satisfaction with tourism to the text. Lines: 516-523.

We thank you for your extraordinary trust,

Authors.

Reviewer 3 Report

This is an interesting paper building well on previous research. The suggestions I make are minor. I would move the Study Area discussion to earlier in the paper, but reduce this significantly. Most readers will not be interested or really understand the value of specific plant species, and will be willing to accept that the areas under discussion are important and should be protected.

Further detail is needed on the sampling and selection of respondents. 

The literature review is reasonable, it would benefit from a more conclusive summary of key points. The use of the "prism" is interesting, the presence of the 4th element, institutional is important, although not as recent as the cited literature suggests, the idea of that element first appeared in Butler, 1974, in Annals of Tourism Research vol2 no 1 100-111, Social Implications of Tourism Development.

The statistical analysis appears sound and is discussed in sufficient detail.

The conclusions are reasonable although there might be some discussion of enlarging their potential application to other areas and types of protected areas/

Overall the standard of English is quite acceptable. A final proof reading would be appropriate.

Author Response

Respected Dear,

we want to thank you for the time you have dedicated to our research! We would like to inform you that we have fully considered all your suggestions. According to each suggestion, we made corrections in the text. We are grateful for the suggestions because they contributed to the quality of this research.

We inform you of the following:

I would move the Study Area discussion to earlier in the paper, but reduce this significantly. Most readers will not be interested or really understand the value of specific plant species, and will be willing to accept that the areas under discussion are important and should be protected.

Answer: we corrected all parts of the text according to your suggestions. We are very grateful for that. Lines: 256-259; 280-286; 301-305.

Further detail is needed on the sampling and selection of respondents. 

Answer: We expanded the text according to your instructions. Lines: 344-421.

The use of the "prism" is interesting, the presence of the 4th element, institutional is important, although not as recent as the cited literature suggests, the idea of that element first appeared in Butler, 1974, in Annals of Tourism Research vol2 no 1 100-111, Social Implications of Tourism Development.

Answer: We especially thank you for this comment, the answer to which will significantly contribute to the quality of the research. We expanded the text according to the instructions. Lines: 224-230.

The conclusions are reasonable although there might be some discussion of enlarging their potential application to other areas and types of protected areas.

The answer. We have expanded the Discussion and Conclusions chapters in everything according to your suggestions. Lines: 583-701.

Thank you once again for your cooperation,

Authors.

Round 2

Reviewer 1 Report

Dear authors,

Thank you for replying to my comments but further improvements are necessary.

Introduction

Comment 1

What is going on in these areas (environmentally, socially and politically)? How has conservation been managed in the past and what policies exist in terms of stakeholder engagement? What issues have emerged in the past? And how do these relate to our broader academic understanding? I would suggest a more in-depth and comprehensive context in your introduction that then raises key questions that your research seeks to address (and this must include something about what influences attitudes and perceptions, since ultimately this is what your study is measuring).

Then in your results and discussion you should focus on the key issues emerging (as you have done), how these relate to the literature elsewhere (as
established in your introduction) and what they mean for local people's engagement in conservation goals in Serbia.

Methodology

Comment 2

The author should be clear on whether the selected communities are found within the PAs OR the selected communities surround the PAs and if they surround the PAs, the author should provide the distance between PAs and selected communities.

Comment 3

And can you say a little more here about how they have engaged with the different  protection/management designations in the past. For example, how were they implemented, were people involved, what has been the relationship since? And involvement in tourism activities in these areas.

Comment 4

You should add something about confidence level and margin of error earlier (so the reader can gauge its representativeness).

Comment 5

Have any data validation procedures been conducted?

Discussion

Comment 6

It is necessary to compare and explain your results, to compare them with actual projects from these areas and from other cases in Europe and around the world. The scale you used is really a lot used in the literature, so it would be nice to make comparisons (not just with your previos research).

Looking forward to reading your manuscript again.

Kind regards,

Reviewer 

The manuscript consists of many grammatical errors which lead to unclear and confusing sentences. Author should avoid the use of brackets and find a way to construct a sentence. Some sentences seem to lose meaning because of the poor command of the English language and use of brackets. Therefore, English should be improved in a revised version. Most sections of the manuscript should be rewritten to improve clarity.

Author Response

Dear reviewer,

we want to inform you that we have taken all your suggestions into account. According to each suggestion, we made corrections in the text. We are grateful for the suggestions because they contributed to the quality of this research.

We inform you of the following:

 Introduction

Comment 1

What is going on in these areas (environmentally, socially and politically)? How has conservation been managed in the past and what policies exist in terms of stakeholder engagement? What issues have emerged in the past? And how do these relate to our broader academic understanding? I would suggest a more in-depth and comprehensive context in your introduction that then raises key questions that your research seeks to address (and this must include something about what influences attitudes and perceptions, since ultimately this is what your study is measuring).

Then in your results and discussion you should focus on the key issues emerging (as you have done), how these relate to the literature elsewhere (as
established in your introduction) and what they mean for local people's engagement in conservation goals in Serbia.

Answer: we made a correction in the article. Lines: 55-64; 121-129.

Methodology

Comment 2

The author should be clear on whether the selected communities are found within the PAs OR the selected communities surround the PAs and if they surround the PAs, the author should provide the distance between PAs and selected communities.

Answer: we have added new text. Lines: 419-431.

Comment 3

 And can you say a little more here about how they have engaged with the different  protection/management designations in the past. For example, how were they implemented, were people involved, what has been the relationship since? And involvement in tourism activities in these areas.

Answer: we have added, lines: 465-484.

Comment 4

 You should add something about confidence level and margin of error earlier (so the reader can gauge its representativeness).

Comment 5

Have any data validation procedures been conducted?

Answer: we have added, lines: 366-373.

Discussion

Comment 6

It is necessary to compare and explain your results, to compare them with actual projects from these areas and from other cases in Europe and around the world. The scale you used is really a lot used in the literature, so it would be nice to make comparisons (not just with your previos research).

Answer: we have added new text. Lines: 645-678.